# Is Thoracic Kyphosis Relevant to Pain, Autonomic Nervous System Function, Disability, and Cervical Sensorimotor Control in Patients with Chronic Nonspecific Neck Pain?

**DOI:** 10.3390/jcm12113707

**Published:** 2023-05-27

**Authors:** Ibrahim M. Moustafa, Tamer Shousha, Ashokan Arumugam, Deed E. Harrison

**Affiliations:** 1Department of Physiotherapy, College of Health Sciences, University of Sharjah, Sharjah 27272, United Arab Emirates; 2Neuromusculoskeletal Rehabilitation Research Group, RIMHS–Research Institute of Medical and Health Sciences, University of Sharjah, Sharjah 27272, United Arab Emirates; 3Faculty of Physical Therapy, Cairo University, Giza 12613, Egypt; 4Sustainable Engineering Asset Management Research Group, RISE-Research Institute of Sciences and Engineering, University of Sharjah, Sharjah 27272, United Arab Emirates; 5CBP Nonprofit (A Spine Research Foundation), Eagle, ID 83616, USA

**Keywords:** thoracic spine, neck pain, kyphosis, sensorimotor control, posture

## Abstract

There is great interest in thoracic kyphosis, as it is thought to be a contributor to neck pain, neck disability, and sensorimotor control measures; however, this has not been completely investigated in treatment or case control studies. This case control design investigated participants with non-specific chronic neck pain. Eighty participants with a defined hyper-kyphosis (>55°) were compared to eighty matched participants with normal thoracic kyphosis (<55°). Participants were matched for age and neck pain duration. Hyper-kyphosis was further categorized into two distinct types: postural kyphosis (PK) and Scheuermann’s kyphosis (SK). Posture measures included formetric thoracic kyphosis and the craniovertebral angle (CVA) to assess forward head posture. Sensorimotor control was assessed by the following measures: smooth pursuit neck torsion test (SPNT), overall stability index (OSI), and left and right rotation repositioning accuracy. A measure of autonomic nervous system function included the amplitude and latency of skin sympathetic response (SSR). Differences in variable measures were examined using the Student’s *t*-test to compare the means of continuous variables between the two groups. One-way ANOVA was used to compare mean values in the three groups: postural kyphosis, Scheuermann’s kyphosis, and normal kyphosis group. Pearson correlation was used to evaluate the relationship between participant’s thoracic kyphosis magnitude (in each group separately and as an entire population) and their CVA, SPNT, OSI, head repositioning accuracy, and SSR latency and amplitude. Hyper-kyphosis participants had a significantly greater neck disability index compared to the normal kyphosis group (*p* < 0.001) with the SK group having greatest disability (*p* < 0.001). Statistically significant differences between the two kyphosis groups and the normal kyphosis group for all the sensorimotor measured variables were identified with the SK group having the most decreased efficiency of the measures in the hyper-kyphosis group, including: SPNT, OSI, and left and right rotation repositioning accuracy. In addition, there was a significant difference in neurophysiological findings for SSR amplitude (entire sample of kyphosis vs. normal kyphosis, *p* < 0.001), but there was no significant difference for SSR latency (*p* = 0.07). The CVA was significantly greater in the hyper-kyphosis group (*p* < 0.001). The magnitude of the thoracic kyphosis correlated with worsening CVA (with the SK group having the smallest CVA; *p* < 0.001) and the magnitude of the decreased efficiency of the sensorimotor control measures and the amplitude and latency of the SSR. The PK group, overall, showed the greatest correlations between thoracic kyphosis and measured variables. Participants with hyper-thoracic kyphosis exhibited abnormal sensorimotor control and autonomic nervous system dysfunction compared to those with normal thoracic kyphosis.

## 1. Introduction

Neck pain is the fourth leading cause of long-term disability with an annual prevalence exceeding 30%, most often in females [1]. Neck pain is a common condition with several proposed biomechanical and psycho-social contributing factors [2]. While the mechanical causes of neck pain are not completely understood, they are thought to be linked to the interconnected functions of anatomical components of the cervical spine [2]. Neck discomfort can be caused by any incident that alters joint mechanics or muscle function via alterations and increases in general loading and load sharing of the various tissues [2,3,4]. For instance, several studies have demonstrated the impact of thoracic spine abnormalities on the kinematics of the cervical spine and overall neck mobility [5,6,7]. In particular, studies have demonstrated a link to movement coordination between the cervical and thoracic spines [3,5,6,8]. While the prevalence of neck disorders is greater in older persons, who also have a higher prevalence of thoracic hyper-kyphosis [6], neck pain is also one of the most common musculoskeletal disorders in young adult populations, with a reported 12-month prevalence ranging from 42 to 67% [9,10,11]. An explanation for such a high rate of neck pain in young and older populations is possible concomitant impairments in the thoracic spine leading to a dysfunction of the cervico-thoracic musculature such as the serratus anterior, levator scapulae, and trapezius [12,13].

Since changes in sagittal thoracic alignment have been reported to alter the mechanical loading of the cervical spine [14,15], this may subtly or overtly impair proprioceptive afferentation from spine ligaments, muscles, and discs, which are considered to be major components of sensorimotor control supplying the essential neurophysiological information for feedforward and feedback responses via linkages to the vestibular, visual, and central nervous systems [16,17,18]. Sensorimotor control is altered in neck pain populations compared to healthy controls, where slower reaction times in visual acuity, cervical movement, and inefficient motor control in general has been reported [19,20]. It is unclear if the altered sensorimotor control is causative of neck pain and disability or a result due to kinesiophobia (fear-based movement variables) [21]; however, it is clear that inefficient sensorimotor control is part of the cycle of chronicity and likely influences recovery [16,17,18,19,20,21]. In addition to sensorimotor control influences, several studies show that the cervical receptors and the sympathetic nervous system have direct interactions [22,23,24]. However, there is limited evidence suggesting that the autonomic nervous system is sensitive to alterations in articular afferent input driven by thoracic hyper-kyphosis and joint dysfunction [22,23,25].

It is known that thoracic hyper-kyphosis is related to a patients’ pain, disability, shoulder kinematics, and general health status [26,27,28,29,30,31]. The threshold for hyper-kyphosis has been reported to be 45° on x-rays (T4-T12 and T5-T12) for pain and disability [26,27], while the 60° value has been reported to be the threshold for more severe disability as in adult spine deformity cases [28,29]. The assumption that a normal thoracic alignment and normal cervical kinematics are important for a better afferentation process has some preliminary evidence [5,6,7,8,12,13,14]. However, studies have not fully investigated the relationship between hyper-kyphosis, forward head posture, and the correlation (if any) on sensorimotor control measurements and the autonomic nervous system.

In general, there is a lack of studies assessing the effect of the thoracic spine sagittal alignment on cervical pain, autonomic nervous system function, disability, and sensorimotor control. Therefore, the purpose of this study was to investigate the correlation in sensorimotor control, neck disability index, and autonomic dysfunction in chronic nonspecific neck patients with a thoracic hyper-kyphosis compared to a matched group of normal kyphosis participants but also having chronic nonspecific neck pain. We hypothesized that patients with chronic neck pain and a thoracic hyper-kyphosis would have impaired sensorimotor control and autonomic dysfunction compared to those chronic neck pain patients with a normal thoracic alignment. Secondarily, we hypothesized that the magnitude of thoracic kyphosis would be correlated to the measures of sensorimotor control and autonomic nervous system function as performed herein.

## 2. Materials and Methods

In this cross-sectional study, we compared 80 young adults over the age of 18 years with chronic nonspecific neck pain and thoracic hyper-kyphosis to 80 matched individuals with chronic nonspecific neck pain who had a normal thoracic kyphotic alignment. Participants were considered matched if their age difference was within 2 years and if their duration of neck pain was of a similar length of time. When the pain duration varied by less than two months, participants were deemed to be matched. Participants were patients recruited from a specialized pain and rehabilitation unit at the Farouk Hospital, Cairo, Egypt from January to August 2022. All cases received a thorough examination in the pain clinic, and all hyper-kyphotic cases underwent radiological assessment. Ethical approval was obtained from the Research and Ethics Committee at Cairo University (CA-REC-22-5-20), with informed consent obtained from all participants prior to data collection in accordance with relevant guidelines and regulations. A flow chart of the recruitment process is shown in Figure 1.

### 2.1. Participant Inclusion and Exclusion Criteria

#### 2.1.1. Inclusion

All participants had to have the diagnosis of chronic non-specific neck pain (CNSNP) with reduced cervical spine range of motion. Thoracic hyper-kyphotic participants were screened with a thorough examination by an orthopedic surgeon, including spine radiography, to rule out serious spine pathologies. However, participants with mild to moderate Scheuermann’s kyphosis (SK) (SK participants were diagnosed via radiography and clinical examination with the orthopedic surgeon) were permitted in the hyper-kyphotic sample, though SK participants were also analyzed as a subgroup of hyper-kyphosis to identify any possible differences. See the results section for details. Participants with normal kyphosis did not receive thoracic spine radiographic imaging, as there was no clinical rationale for imaging in these participants; thus, an external measurement of thoracic kyphosis was chosen to make comparisons in all participants. Prior to inclusion, participants were evaluated by measuring the sagittal thoracic *kyphotic angle ICT-ITL (max)* using the 4D formetric system (note it is a 4D system, as it allows for a time variable to capture any sagittal shift and sway over 60 s) where ICT-ITL (max) is measured between tangents from the cervicothoracic junction (ICT-T1) and that of the thoracolumbar junction (ITL-T12). The reproducibility of results is excellent, making this non-invasive system appropriate for clinical assessment, as the reliability of thoracic kyphosis measurement is excellent with coefficients of variation of approximately 7% (3.5 degrees) for angulations [32,33]. Figure 2 depicts this measurement. Hyper-kyphosis participants were included if the *ICT-*ITL** (*max*) angle measured more than 55°. Normal kyphosis participants were defined as the ICT-ITL (max) angle being less than 55° [33]. There is good correlation between the formetric vs. Cobb angle of thoracic kyphosis, but formetric measurements consistently overestimates kyphosis by an average of 5–7°, indicating that the radiographic kyphosis would be approximately 48–50°, which is the upper end of normal and the cutoff value for where thoracic kyphosis begins to be associated with pain and disability [26,27,30,31,33,34,35].

#### 2.1.2. Exclusion

Exclusion criteria included the presence of any signs or symptoms of medical “red flags”, a history of previous spine surgery, vertebral fracture, signs or symptoms of upper motor neuron disease, vertebrobasilar insufficiency, amyotrophic lateral sclerosis, and bilateral upper extremity radicular symptoms. Detailed exclusions were:Neck pain associated with whiplash injury;Neck pain with bilateral cervical radiculopathy;Fibromyalgia syndrome;Surgery in the neck area, regardless of the cause;Neck pain accompanied by vertigo caused by vertebra-basilar insufficiency or accompanied with non-cervicogenic headaches;Recent or recurrent middle ear infections or any hearing impairment requiring the use of a hearing aid;Visual impairment not corrected by glasses;Any disorder of the central nervous system.

### 2.2. Measurement Procedures

#### 2.2.1. ICT-ITL (Max)

The thoracic posture was measured in a neutral position to ensure consistency between repeated images captured in the same session; also, this would aid comparison with other studies that measured Cobb’s angle for thoracic kyphosis in radiographic studies. Each participant was positioned 2 m from the measurement system in front of a black background screen, and a valid and reliable formetric system [32,33,35] was used to analyze 3D body posture displacements (DIERS Medical Systems, Chicago, IL, USA). The column height was aligned to move the relevant parts of the patient’s back into the center of the control monitor by using the column up/down button of the control unit. A permanent mark fixed with a tape on the floor was used to ensure the best lateral and longitudinal position of the patient. The participant’s back (including the upper gluteal region) was uncovered to allow better imaging of the back. The participants’ hair was tied up (when needed) to allow visualization of the vertebral prominences. The system was ready for image recording when the participant was correctly positioned in the participant’s perception of their neutral resting, relaxed posture position, being defined as the relaxed upright stance, with feet hip width apart and barefooted, where the participant was instructed to:look straight ahead in a relaxed breathing state with their head in a neutral position, not being twisted or bent;relax their shoulders, do not hunch them or rotate them forward;keep their upper arms, elbows and hands comfortably at their sides;stand with their legs straight, but with knees relaxed, not locked back (preventing hyperextension).

Thoracic kyphosis was measured as the maximum kyphosis between tangents from the cervicothoracic junction (ICT-T1) and that of the thoracolumbar junction (ITL-T12). This would be considered a total thoracic kyphosis from T1–T12 vertebral levels. Kyphotic participants were included if the angle measured 55° or more and normal kyphosis if the angle measured less than 55° [26,27,30,33,34,35]. There is a good correlation between the formetric measurement and Cobb angle of thoracic kyphosis, but the former one consistently overestimates kyphosis by an average of 5–7° [33,35].

#### 2.2.2. Craniovertebral Angle (CVA)

To assess the influence of thoracic kyphosis on forward head posture (FHP), we measured the craniovertebral angle (CVA) in both groups. The CVA is constructed using C7 spinous process and drawing a line from it to the tragus of the ear. Next, a horizontal line is drawn through C7 spinous, where the CVA is the acute angle between the two lines. Typically, when the CVA is less than 50°, then a participant is classified as having significant forward head posture [36,37]. The CVA has excellent reliability to assess forward head posture [36,37]. Figure 3 presents the CVA.

#### 2.2.3. Numerical Rating Score (NRS)

Neck pain average intensity over the previous week was assessed using a 0–10 NRS score ranging from 0 = no pain to 10 = bed ridden and incapacitated. The reliability and validity of the NRS has been found to be good to high [38].

#### 2.2.4. Neck Disability Index

The neck disability index (NDI) to assess activities of daily living impact was administered. The NDI has good reliability, validity, and responsiveness to change [39].

#### 2.2.5. Sensorimotor Control Measures

There is a detailed interplay between proprioception and postural control, such that normal posture alignment is likely a major component driving the afferentation process leading to improved sensorimotor integration and motor control. To assess the effects of thoracic kyphosis and forward head posture on the sensorimotor system, we measured three common measures of sensorimotor control herein, including the assessment of the following: (a) cervical joint position sense testing, (b) head and eye movement control, and (c) evaluation of postural stability.

a.Cervical Joint Position Sense Testing

Head repositioning accuracy (HRA) was assessed with the cervical range of motion (CROM) device as previously described in the literature (CROM deluxe device by Frabication: https://www.amazon.com/Fabrication-12-1156-Crom-Deluxe/dp/B00BRCGCNO, accessed on 19 May 2023). We followed the protocol of Loudon et al., as this is reliable and valid [40]. The CROM was placed on the participants’ head while they were seated upright on a stool without a backrest, with both feet supported on the floor with knees flexed to ≈90°. The participant was asked to sit upright in a neutral, non-slouched, and comfortable thoracic posture attempting to keep the thoracic spine perpendicular to the plane of the stool. The neutral head position (NHP) was considered as the starting and reference position, where the CROM was adjusted to zero for the primary plane of rotational movement. Patients were instructed to close their eyes, memorize the starting position, actively rotate their head to 30° about the vertical axis, and reposition their head to the starting position with no restrictions for speed; only repositioning accuracy was encouraged. HRA was defined as the difference in degrees between the starting and the return positions [41]. Three repetitions were performed within 60 s for both the left and right directions; for a total of six sets. The test is reported as error in degrees (°), where less than 10% or 3° is normal [40,41].

b.Head and eye movement control: smooth pursuit neck torsion test (SPNT)

Assessment of disturbances in eye movement control by the electro-oculography was adopted from Tjell et al. [42]. The test was performed with the participant’s head and trunk in a neutral straight ahead position and then two trunk rotation positions (head neutral, trunk in 45° rotation to each side). Patients were asked to blink three times (for recognition and elimination in data analysis) and then follow the path of a light as closely as possible with their eyes. The SPNT test value was defined as the difference between the average gain in the neutral and torsion positions for left vs. right rotation. Findings are reported as a percentage (%) of error of corrective saccades (eye movements), where 100% is perfect (0% error), 10–20% error is normal, and greater than 20% error is abnormal. The videonystagmography system VisualEyes™ 525 by Interacoustics A/S in Denmark was utilized to conduct the SPNT test.

c.Postural stability

The Biodex Balance System SD (Biodex Medical Systems, Inc., Shirley, NY, USA) was used to assess postural stability. Dynamic balance was assessed by simulating displacements in both anterior/posterior (AP) and medial/lateral (ML) directions by changing the device platform level of stability. The platform provides an objective assessment of balance using three indices: the overall stability index (OSI), an anteroposterior stability index (APSI), and a mediolateral stability index (MLSI). These indices are calculated according to the degree of platform oscillation. Smaller values indicate a good stability level of the participants. The reported inter-examiner reliability coefficients range between 0.77 and 0.99 [43,44]. Balance indices were calculated over three 10 s trials, with 20 s rest between trials. The average of three trials was recorded. The balance system was set to a dynamic position of 4 out of 8.

#### 2.2.6. Sympathetic Skin Response (SSR)

On the day of the study, patients were asked to avoid using medicated lotions and cosmetics (on the hands), not to engage in physical activity, and avoid smoking, eating, and drinking coffee two hours prior to the recordings. To acclimatize patients to the experimental environment, all participants spent 20 min in a room with a temperature of 22–24 °C just before the measurements were taken.

The EMG was used to measure the SSR. Room temperature was maintained at 26 °C in order to maintain a stable skin temperature [45,46]. The active surface electrodes were attached on the palmar side, and the references were placed on the dorsum of the hand. The stimulus was given at the wrist contralateral to the recording side. Measurements were taken from both left and right sides. An intensity of 20–30 mA with an irregular interval of more than one minute was applied to prevent habituation. When habituation occurred, stimulation was delayed for about three or four minutes. Skin potentials were recorded for a 10 s analysis period. The latency and peak-to-peak amplitude SSR were determined. Mean values of three trials were used for each parameter. Sweep speed was 500 ms/div.

SSR was considered absent if there was no response after 10 stimuli [47]. In the SSR trace, the latency and amplitude character points markers placement was corrected manually if the ones automatically generated by the EMG software were inaccurately placed. Latencies were measured from the stimulation artifact to the initiation of the response which is defined as the earliest point where the amplitude begins to increase. The amplitude is measured from the peak of the first deflection to the peak of the next one (peak-to-peak) [48].

### 2.3. Sample Size Determination

A priori sample size calculation based on a pilot study conducted for 10 patients indicated that 70 participants per each group would be required to detect an effect size of 0.6. Pain was used as the outcome measure for this calculation. To insure robust data, the sample size was increased by 14% in order to attain 80 participants per group.

### 2.4. Statistical Analysis

The one-sample Kolmogorov–Smirnov normality test was used to determine whether the data were normally distributed, and homogeneity of variance assumption was assessed by the Levene statistic. Descriptive data were presented as mean ± standard deviation. The Student’s *t*-test was used to compare the means of continuous variables, and the Chi-squared test for categorical variables was used to assess any differences between the two groups, the entire hyperkyphotic and normal groups. When separating the hyper-kyphosis sample into the two subgroups, the one-way ANOVA was used to compare the mean values in the three groups: postural kyphosis, Scheuermann’s kyphosis, and normal kyphosis group. Post hoc Tukey’s analysis was performed to determine differences between groups, when ANOVA revealed a significant difference.

A *p*-value < 0.05 was considered statistically significant. Correlations (Pearson’s *r*) were used to examine the relationships between the ICT-ITL (KA-max) in both groups and the measured variables: SSR amplitude and latency, OSI, left and right rotation repositioning accuracy, NDI, SPNT, and NRS. The minimal clinically important difference (MCID) of the of the SSR and NDI outcomes were compared to the existing literature [45,46]. Whereas the MCID of the sensorimotor control variables were not available in the literature to our knowledge thus, effect sizes for all variables were measured using Cohen’s d, where d ≈ 0.2 is limited effect, d ≈ 0.5 is a moderate effect, and d ≈ 0.8 is a large effect with very significant clinical relevance. Correlations were investigated for each group (postural kyphosis, Scheuermann’s kyphosis, and normal kyphosis) separately and then as an entire sample of 160 participants to identify possible differences. SPSS version 20.0 software (SPSS Inc., Chicago, IL, USA) was used for analyzing data with normality and equal variance assumptions ensured before the analysis.

## 3. Results

### 3.1. Participant Demographics and Characteristics

Descriptive data for the demographic and clinical variables for the entire sample of 80 hyper-kyphotic and the 80 normal kyphosis participants are presented in Table 1. No statistically significant differences between the hyper-kyphotic group and the normal kyphosis group were found at baseline for their demographic and clinical variables. No data were missing for any of measured variables in any of the participants in this study. We separated the hyper-kyphotic participants into two groups: 35 postural kyphosis and 45 Scheuermann’s kyphosis categories, and Table 2 presents this demographic and clinical data. No statistically significant baseline differences for the clinical and demographic variables was found for these two subgroups of thoracic hyper-kyphosis.

### 3.2. Between Group Analysis

#### 3.2.1. ICT-ITL (Max)

Box and whisker plots of the ICT-ITL (max) in the two hyper-kyphotic groups compared to the normal group are presented in Figure 4. As designed by our inclusion criteria, both hyper-kyphotic groups had the largest ICT-ITL (max) angles indicating an exaggerated kyphotic posture (entire hyper-kyphotic group, 67° ± 4; postural kyphosis group, 66.5° ± 3; and Scheuermann’s kyphosis group, 67.5° ± 4.9). The normal kyphosis group had the smallest ICT-ITL (max) angles (normal kyphosis, 49° ± 3). As can be seen in Figure 4, there was no overlap between the kyphotic angles of the normal and kyphotic groups. Those with thoracic hyper-kyphosis were well above the threshold of 55°, thus eliminating any overlap within the standard error of measurement of the formetric system.

#### 3.2.2. NRS and NDI

For pain level on the NRS, we found no statistically significant differences in pain intensity between groups (*p* > 0.05). However, the entire sample of the hyper-kyphotic group showed an increase in neck disability (NDI) scores compared to the normal kyphosis group (*p* < 0.001). When separating the hyper-kyphosis sample into the two subgroups, we identified a statistically significant difference in the NDI, where the Scheuermann’s kyphosis group had a higher disability. Table 3 and Table 4 presents these results.

#### 3.2.3. Sensorimotor Control Variables

The unpaired *t*-test analysis showed that there were statistically significant differences in the hyper-kyphotic group versus the normal kyphosis group for the sensorimotor control variables. For OSI, we found significant abnormality (less stability) in dynamic stability for the hyper-kyphotic group compared to the normal kyphosis group (*p* < 0.001); smaller values indicate a good stability level of the participants. Larger errors were evident for right and left rotation repositioning accuracy (*p* < 0.001) in the hyper-kyphotic group as well; results are reported as error in degrees (°) where less than 10% or 3° is normal. For SPNT, we found a significant difference between the two groups with a larger average gain for the hyper-kyphotic group; results are reported as a percentage (%) of error of corrective saccades, where 100% is perfect (0% error), 10–20% error is normal, and greater than 20% error is abnormal. Table 5 presents this data.

Between group comparisons for the postural kyphosis, Scheuermann’s kyphosis and normal groups are presented separately for sensorimotor control variables and the CVA in Table 6. Overall, the Scheuermann’s kyphosis group is shown to have statistically and clinically significant worse sensorimotor control variables. Similarly, the Scheuermann’s kyphosis group has a statistically significant reduction in the CVA indicating more forward head posture; *p* < 0.001, Table 6.

#### 3.2.4. SSR Latency and Amplitude

For neurophysiological variables, we found an increase in SSR amplitude in the entire hyper-kyphotic group compared to the normal kyphosis group (*p* < 0.001). In contrast, no such difference was evident for in SSR latency (*p* = 0.07) as presented in Table 5. Between group comparisons for the postural kyphosis, Scheuermann’s kyphosis, and normal groups are presented for SSR latency and amplitude in Table 6. SSR data show a statistically significant increased amplitude and a faster latency for the Scheuermann’s kyphosis; however, the latency difference is a rather weak clinically and non-significant (effect size 0.2; *p* = 0.29). See Table 5.

### 3.3. Correlations

Pearson r correlations between the magnitude of thoracic kyphosis are presented in Table 7 for both subgroups of thoracic hyper-kyphosis, the normal kyphosis group, and the entire sample of 160 participants. The kyphotic angle showed a moderate positive correlation for all sensorimotor control variables (SPNT, OSI, and right and left rotation repositioning accuracy) with the postural kyphosis group showing significantly greater correlations than the other groups. We found a moderate positive correlation between the thoracic kyphotic angle and SSR amplitude for the entire sample of 180 participants (*r =* 0.69, *p* < 0.001), indicating as the kyphotic angles increased, the SSR amplitude increased in our population. Again, the strongest correlation was found for the postural kyphosis group. In contrast, we found a low negative correlation between the kyphotic angle and SSR latency for the entire sample of 180 participants (*r =* −0.49, *p* < 0.001), with the smallest correlation found in the postural kyphosis group. Additionally, pain and disability scores were moderately linearly correlated to the magnitude of kyphosis in the entire sample (NRS: *r* = 0.53, *p* < 0.001; NDI: r = 0.67; *p* < 0.001) with the postural kyphosis group showing slightly stronger correlations than the other participants. Table 7 presents this data in detail.

#### Craniovertebral Angle (CVA)

Box and whisker plots of the CVA in both hyper-kyphosis groups (postural kyphosis and Scheuermann’s kyphosis) and the normal kyphosis group are presented in Figure 5. Overall, the Scheuermann’s kyphosis group had the smallest CVA indicating greater forward head posture than the other two groups; CVA 38.5° ± 4.5. The normal kyphosis group had the greatest CVA indicating a more neutral sagittal head posture; CVA 53° ± 4. These results were statistically significant (*p* < 0.001). Lastly, the CVA is negatively correlated with the magnitude of thoracic kyphosis in all groups, with the strongest correlation found in the posture kyphosis group, indicating that as the magnitude of thoracic kyphosis increases, the CVA decreases and forward head posture increases (entire sample r = −0.061, *p* < 0.001). See Table 7.

## 4. Discussion

The results of the current study demonstrate that the sensorimotor control, disability, and autonomic nervous system function of patients with chronic nonspecific neck pain and thoracic kyphosis are distinctly different compared to those patients with normal thoracic alignment. Thus, our study’s primary hypotheses are confirmed by these findings. As far as we know, this is the first study to provide objective evidence that these specific outcomes are differently affected by altered sagittal thoracic alignment. These differences cannot be explained in the context of the proposed different pain intensity or pain duration differences among groups, as the between group analysis revealed a non-significant difference between groups for both these variables. Most importantly, the difference between groups appear of clinical importance, as reflected by their effect sizes (d > 0.5) and the mean differences between groups, which are greater than the minimal clinically important difference (2.77 × SEM) for the SSR and the NDI outcomes [49,50,51].

### 4.1. Thoracic Kyphosis

Thoracic hyper-kyphosis represents one of the top four spine abnormalities associated with adult spine deformity (ASD), a world-wide, known set of spine deformities and associated disabilities affecting adults over the age of 18 years [28,29]. For example, Pellise et al. [28] identified that patients with radiographically determined thoracic hyper-kyphosis ≥60° had significantly lower health-related quality of life scores compared to patients afflicted with four other major health disorders (type II diabetes, rheumatoid arthritis, heart disease, or pulmonary disease). There are currently different proposed cut-off values that distinguish between normal and hyper-kyphosis. For example, 50° is suggested by some studies as a cut-point for thoracic hyper-kyphosis [30,31], while other investigations have identified that the cut-point between those with pain, lower self-image, and decreased function is 45° [26,27,52]. In the current investigation, we used a 4-D formetric scanner to evaluate the external measurement of thoracic kyphosis, and in the hyper-kyphosis group our average participants’ kyphosis was 67°, while it was 49° in the normal kyphosis group. For comparison, it is known that the formetric and inclinometry measures of external thoracic kyphosis overestimate the radiographic determined thoracic kyphosis by approximately 5–7° and maybe more depending on the unique patient population [33,35,53,54]. Using this information, we estimate that our hyper-kyphosis group had a radiographic measured thoracic kyphosis averaging 60° (depending on the vertebral levels of measurement) meaning that this group would be at the threshold for ASD and that they would certainly be classified as an abnormal spine deformity group [28,29].

### 4.2. CVA, Pain, Disability, and Sensorimotor Control

In Table 7, we separated our study’s findings into four separate correlation analyses: postural kyphosis, Scheuermann’s kyphosis, normal kyphosis, and the entire population. This was chosen due to the possibility of identifying a stronger correlation between a specific variable within the hyper-kyphosis groups compared to the normal group. In this regard, most variables showed stronger correlations within the postural hyper-kyphosis group compared to the other two populations. It is unclear what this means in terms of chronic neck pain and neck disability in our study, but it may prove significant in future investigations. Between group differences in sensorimotor control and neck disability scores were identified, while there were no differences in pain intensity and duration between groups. The relationship between pain intensity and thoracic alignment has been detailed in several studies, where some investigations have reported significant positive associations, while other studies demonstrated no association between the two variables [14,55,56,57]. One such investigation concluded that neck pain was positively associated with hyper-kyphosis during a functional typing task [58]. These conflicting results might be due to multiple factors, such as the severity of chronic pain determined by a variety of other physical and psychosocial contributing factors [59]. Therefore, it is difficult to predict any linear relationship between thoracic kyphosis and neck pain intensity. Since the differences in disability and sensorimotor control found between our hyper-kyphosis and normal groups are not due to differences in pain intensity or pain duration, we propose the possible mechanism driving these changes might be dysafferentation mediated by abnormal forward head malalignment and increased thoracic kyphosis.

Increased thoracic kyphosis leads to the anterior shift of the trunk mass through an alteration of the thoracic spine loading, thereby resulting in forward head posture of the cervical spine as a direct compensation [14]. This has been confirmed in the current study by the fact that the mean CVA for the kyphotic group was significantly lower than that of the control (non-kyphotic) group indicating considerably larger forward head posture in the kyphotic group. Sustained forward head posture is implicated in the alteration of cervical motor control and the development of myofascial dysfunction. The assumption that abnormal forward head posture alignment is important for the afferentation process has some preliminary evidence. For instance, two modeling studies have predicted that as forward head posture increases, increased stress and strain are placed upon the muscles and ligaments of the cervical and thoracic region [60,61,62]. Increased forward head posture results in altered cervical spine alignment and shoulder joint position, causing abnormal kinematics and neurophysiologic afferent input (the so-called dysafferentation) [63,64,65]. We suggest that this information is consistent with and may partially explain the findings from Stanton and colleagues [66], where chronic idiopathic neck populations were identified to have an abnormal ability to return the cervical spine to the neutral position (altered sensorimotor control).

In the current investigation, it is difficult to discern between the effects of increased forward head posture (the CVA) versus increased thoracic kyphosis on the variables we have assessed, and conflicting results have been reported in the literature regarding the significance of sensorimotor control measures in neck pain populations. For example, in a recent systematic review with meta-analysis, it was found that increased forward head posture is associated with the presence of neck pain in adults [67,68]. However, Pacheco and colleagues [68] found that forward head posture was not different between young collegiate adults with ”subclinical neck pain” compared to asymptomatic controls. This later investigation [69] used a very different participant population in both age (college students only 18–22 years) and a non-clinically relevant pain condition (treatment was not sought) as compared to our current investigation (significantly older and participants were seeking intervention from our pain clinic); thus, we believe our results to be more in line with the two recent systematic reviews with meta-analysis [67,68].

A significant negative correlation was found in the current study between the magnitude of thoracic kyphosis and a participant’s CVA. This finding was previously reported in the study by Quek et al. [13]. Moreover, a multitude of biomechanics analysis have revealed that increased forward head posture along with thoracic hyper-kyphosis is associated with mobility limitations in the cervical spine [15]. Given the preliminary evidence for the significant role of normal sagittal configuration in normalizing the afferentation processes, it is not surprising that there was a considerable between group difference in the sensorimotor control variables. The current study’s findings of increased disability and more disturbed sensorimotor control add credence to the above biomechanics and clinical investigations detailing the effects of thoracic spine abnormalities on the cervical spine. The relationship between increased forward head posture, that is, a smaller CVA, and thoracic kyphosis has been investigated in previous studies [14,15]. Lau et al. [14] reported a smaller CVA in participants with neck pain compared to a healthy control group. Lau et al. [14] suggested that a smaller CVA and upper thoracic angle were thought to be predictors of neck pain and disability in terms of their participants pain intensity. However, in the current study, it was not surprising that there were no significant variations in pain intensity between our two groups, because pain is a multidimensional phenomenon affected by many factors other than sagittal alignment. Moreover, symptoms caused by abnormal spine biomechanics likely appear after the consequences of mechanical distortions have progressed to the point where the body’s adaptive ability has been overcome (as is the case with heart disease, cancer, hypertension, etc.). Since the participants in our study were much younger than those in the other study [14], the age differences between the two studies could explain the disparity in pain intensity findings. Interestingly, although the different postural alignments between our groups had no effect on pain intensity, it had a significant impact on the other measurement outcomes, as shown by the strong correlation between spinal alignment and those outcomes (disability, sensorimotor control measures, and sympathetic skin resistance). Our finding is consistent with that of Moustafa et al., who found that even in asymptomatic individuals with a forward head posture, there are significant abnormal neurophysiological responses, including prolonged central conduction time and abnormal sensorimotor integration [70].

### 4.3. SSR

Our choice of the sympathetic skin response (SSR) as an indicator for autonomic nervous system (ANS) function in the current study instead of other measures such as heart rate variability (HRV) measurement might by questioned. HRV is a commonly used and standardized method for assessing ANS function, as it provides separate metrics for sympathetic and parasympathetic functions through the low-frequency (LF) and high-frequency (HF) spectral components of HRV. However, recent studies have shown that the traditional HRV framework established in the 1980s has limitations in dealing with the evidence accumulated over the past half-century. As pointed out by Hayano and Yuda [71], using HRV without criticism may lead to incorrect conclusions or judgments. Moreover, a study by Ke et al. [72] has shown that both SSR and HRV parameters are sensitive in determining ANS dysfunction. Therefore, we chose SSR as an alternative and easily assessed measure for ANS function in our study. We acknowledge that HRV may provide additional information about ANS function, and future studies should use this to assess the influence of thoracic kyphosis and increasing FHP on HRV. However, our current findings using SSR highlight the potential clinical value of this measure in assessing ANS dysfunction.

We believe that a significant between group difference in SSR indicates the considerable role of spinal sagittal alignment in maintaining the normal function of the autonomic nervous system. Oakley et al. [73] detailed information indicating that restoring normal posture and spine alignment has important influences on neurophysiology, sensorimotor control, and autonomic nervous system function. There is limited but high-quality research identifying that sagittal spine alignment restoration plays an important role in improving neurophysiology, sensorimotor control, and autonomic nervous system function [73,74]. Disturbances in the afferentation process may be the possible explanation underpinning spine-related autonomic dysfunction. An adverse mechanical tension acting on the brainstem and cranial nerves 5–12, specially the 10th cranial nerve, may be one of the fundamental mechanisms that explain the autonomic dysfunction in the kyphotic group compared to the control group.

### 4.4. Clinical Relevance

Clinically, our study findings would implicate the thoracic hyper-kyphosis as a contributing factor in the disability levels reported in chronic non-specific neck pain disorders. We identified that increased FHP (a decreased CVA) is corelated to the magnitude of thoracic kyphosis. Since it is known that increasing FHP causes a simultaneous increased loading of the upper thoracic and lower cervical spine, it would be logical that this increased loading affects the ability of a person’s cervical spine to perform complex and simple tasks that create further functional demands on the spine tissues [60,61]. Furthermore, increased FHP alters both the total range of motion and segmental kinematics of the cervical spine during movements, and this would further exacerbate cervical spine pain and create limits to functional movements as a result [13,14,15]. Similarly, the general results of our sensorimotor control assessments indicate that participants with increased thoracic kyphosis have a generally poorer ability to perform efficient tasks requiring stability (balance), movement accuracy (HRA), and ocular motor control (SPENT). The findings of inefficient sensorimotor control would have significant implications for continued injury (increased and altered stresses and strains on various spine tissues) of a participants cervical spine tissues, where a vicious cycle is set up of spine tissue damage due to inefficient motor control or coordination of movement. In general, our findings would suggest that structural rehabilitation (rehabilitation aimed at improving spine alignment) of the hyper-kyphotic spine should be a primary goal of patient treatment procedures. In fact, in a recent randomized trial, it was identified that structural rehabilitation of the thoracic hyper-kyphosis had positive effects on improving chronic non-specific neck pain, disability, and sensorimotor control as compared to standard rehabilitative care that did not improve the alignment of the thoracic hyper-kyphosis [74].

### 4.5. Limitations

The current study has limitations to consider which should lead to future investigations. First, the outcome measures used to verify if thoracic kyphosis affects sensorimotor control, pain, and disability may not be the only ones or the ideal assessment for chronic neck pain outcomes. Additionally, we measured the thoracic kyphosis using an external posture assessment device, and this does not provide the same quantitative data as radiographic or other imaging methods used for the measurement of thoracic kyphosis. Similarly, although the CVA is a valid and reliable method for measuring forward head alignment [14,36,37], it might not adequately describe the actual sagittal cervical vertebral alignment. Using the sagittal radiological profile would thus give further insights into exact rotation and translation displacements of individual vertebral and overall cervical curvature geometry and magnitude. Furthermore, our study did not include a true normal control group without chronic non-specific neck pain and a normal thoracic kyphosis; thus, comparison to populations without chronic non-specific neck pain cannot be made. Finally, although we demonstrated that increasing kyphotic magnitudes of the thoracic spine are correlated with sensorimotor control measurements and the autonomic nervous system function, it must be emphasized that correlation does not imply causation. Future investigations that are prospective and longitudinal in design along with randomized interventional trials are needed to confirm the relationship between the magnitude of thoracic hyper-kyphosis and the measures reported herein.

## 5. Conclusions

This case control on a chronic non-specific neck pain population identified that those with thoracic hyper-kyphosis also have an increased forward head posture (reduced CVA) and that this is related to abnormal autonomic nervous system function. Furthermore, increased thoracic kyphosis is correlated to disturbances of a variety of sensorimotor control measures. Our findings may have important implications for the assessment and rehabilitation of these populations of patients with hyper-kyphosis of the thoracic spine, increased forward head posture, and chronic non-specific neck pain.

## Figures and Tables

**Figure 1 jcm-12-03707-f001:**
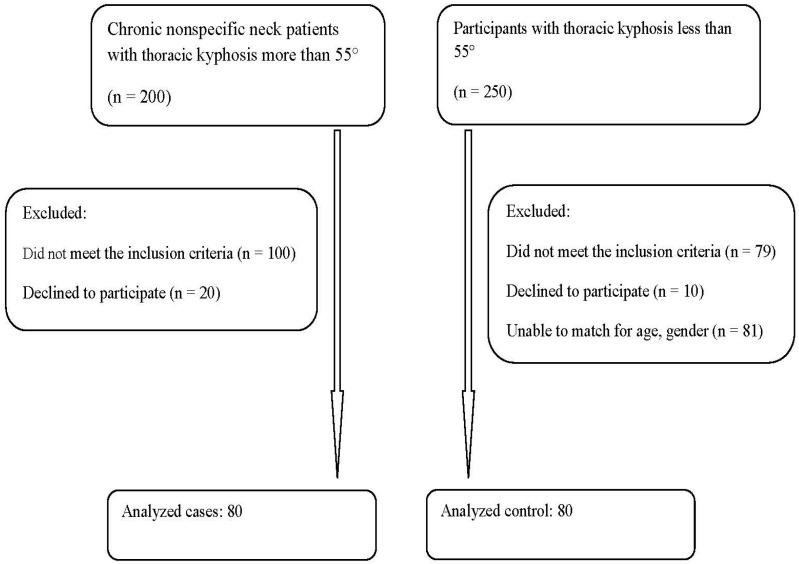
Participant study flow chart for group inclusion and exclusion.

**Figure 2 jcm-12-03707-f002:**
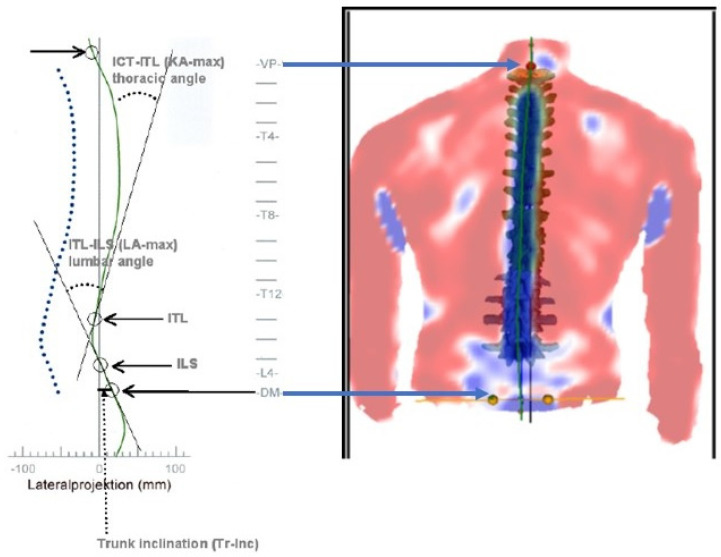
4D formetric device measurement of thoracic kyphosis and trunk inclination where *kyphotic angle ICT*-*ITL* (*max*) is measured between tangents of cervicothoracic junction (ICT) and of thoracolumbar junction (ITL). ICT: inflectional points from cervical to thoracic spine. ITL: inflectional points from thoracic to lumbar spine. KA: kyphosis angle. LA: lordosis angle. VP: vertebra prominence. DM: dimple.

**Figure 3 jcm-12-03707-f003:**
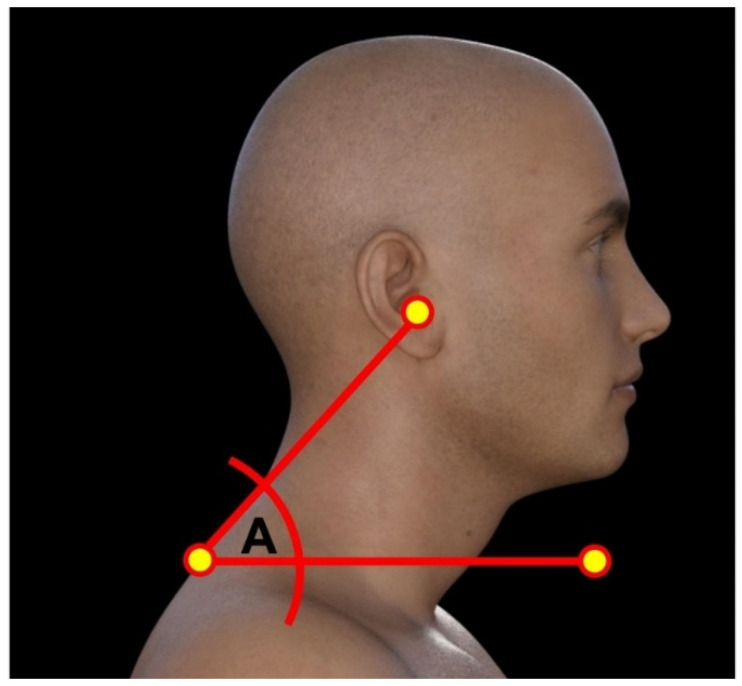
Measurement of the craniovertebral angle (CVA). Two markers are utilized and placed at the level of the C7 spinous process and the tragus of the ear; then a line is constructed connecting these two points. Finally, a horizontal line is drawn using the C7 marker as the reference, and the CVA is measured as angle A between the two lines [36,37].

**Figure 4 jcm-12-03707-f004:**
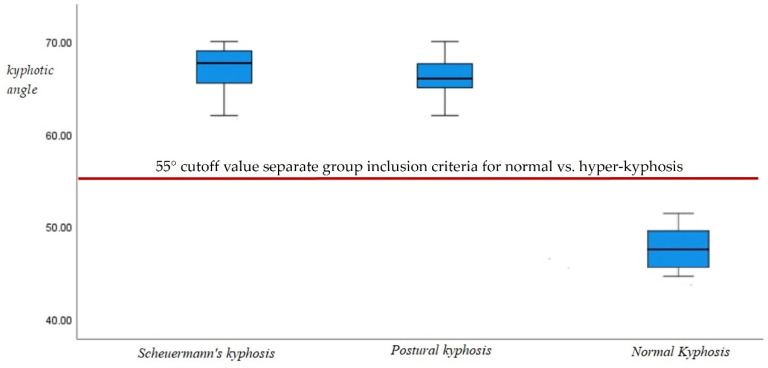
Box and whisker plots shown of the magnitude of thoracic kyphosis, ICT-ITL (max), in both hyper-kyphotic groups (postural kyphosis, 66.5° ± 3; Scheuermann’s kyphosis; 67.5° ± 4.9) and the normal kyphosis (49° ± 3) groups. A statistically significant difference for these variables between normal kyphosis and total hyper-kyphosis (but not for hyper-kyphosis type) was forced by study design, where 55° (shown as red-dashed line) was the absolute cutoff for kyphosis between groups.

**Figure 5 jcm-12-03707-f005:**
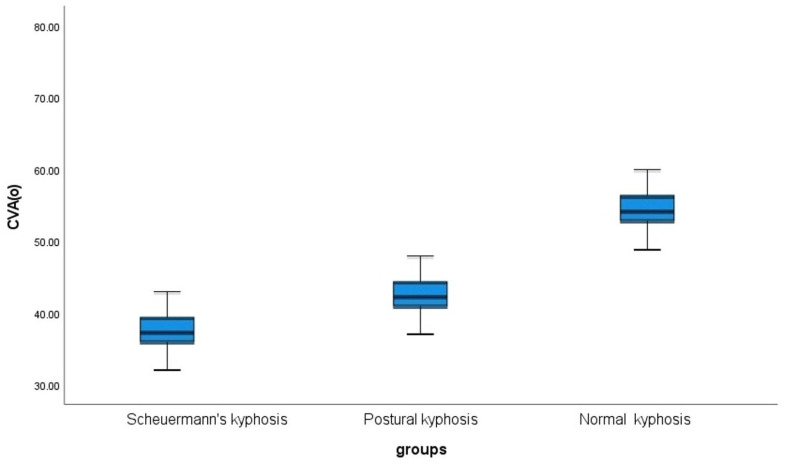
Box and whisker plots of the craniovertebral angle measured in degrees (CVA°) in the postural kyphosis groups (CVA, 44° ± 4), the Scheuermann’s kyphosis group (CVA, 38° ± 4.5), and the normal kyphosis (CVA, 53° ± 4) groups.

**Table 1 jcm-12-03707-t001:** Baseline participant demographics. The statistical significance between groups is shown. Here both the postural and Scheuermann’s kyphosis group are combined into an entire kyphotic sample. The Student’s *t*-test to compare the continuous variables and the Chi-squared test for categorical variables were used. Values are expressed as means ± standard deviation where indicated.

Variables	Entire Kyphotic (n = 80)	Normal (n = 80)	*p* Value
Age (years)	25.1 ± 3	24 ± 4.6	0.07
Weight (kg)	66 ± 10	60 ± 9	0.9
**Sex**			
Male	38	32	0.2
Female	42	48	
**Marital status**			
Single	61	59	0.3
Married	19	21	
Separated, divorced, or widowed	0	0	
Pain duration (months)	18 ± 4	17 ± 5	0.16
**Smoking**			
Light smoker	29	32	0.4
Heavy smoker	14	15	
No Smoker	37	33	

**Table 2 jcm-12-03707-t002:** Participant demographics of the hyper-kyphotic group separated by type of kyphosis with either a postural kyphosis or a Scheuermann’s kyphosis. Statistical significance was tested using the ANOVA test to compare continuous variables, and the Chi-squared test for categorical variables. Values are expressed as means ± standard deviation. * is a statistically significant difference.

Variables	Postural Kyphosis N = 35	*Scheuermann’s kyphosis* *N = 45*	Normal (n = 80)	*p* Value
Age (years)	25 ± 3.2	25.3 ± 3	24 ± 4.6	0.16
Weight (kg)	65 ± 11	67 ± 9	60 ± 9	0.6
**Sex**				
Male	18	20	32	0.5
Female	17	25	48	
**Marital status**				
Single	27	33	59	0.6
Married	8	12	21	
Separated, divorced, or widowed	0	0	0	
Pain duration (months)	17 ± 3	18.7 ± 4.5	17 ± 5	0.1
**Smoking**				
Light smoker	15	14	32	0.15
Heavy smoker	8	6	15	
No Smoker	12	25	33	
Kyphotic angle	66.5 ± 3	67.5 ± 4.9	49 ± 3	<0.001 *

**Table 3 jcm-12-03707-t003:** Between-group comparisons of pain and disability outcomes.

Variables	Entire Kyphotic Group (n = 80)	Normal Group (n = 80)	Cohen’s d Effect Size	*p* Value (95% CI)
NDI	37.3 ± 4.1	29.8 ± 2.4	2.2	<0.001 * [−8.5, −6.45]
Pain intensity	5.3 ± 2.0	4.9 ± 1.8	0.20	0.18 [−0.99, 0.19]

CI = confidence interval; NDI = neck disability index; Pain intensity is 0–10 where 0 is no pain and 10 is incapacitated; all values are expressed as means ± standard deviation. * = statistically significant.

**Table 4 jcm-12-03707-t004:** Results of one-way-ANOVA and post hoc (Tukey) test. * = statistically significant.

	Postural Kyphosis N = 35	Scheuermann’s Kyphosis N = 45	Normal Group (n = 80)	F-Value/ *p*-Value	Post Hoc
NDI	35.2 ± 2.4	39.1 ± 4.5	29.8 ± 2.4	132.67/ <0.001 *	Group 1 vs. Group 2: Diff = 3.9, 95% CI = 2.22 to 5.57, *p* < 0.001 * Group 1 vs. Group 3: Diff = −5.4, 95% CI = −6.90 to −3.89, *p* < 0.001 * Group 2 vs. Group 3: Diff = −9.3, 95% CI = −10.68 to −7.91, *p* < 0.001 *
Pain intensity	4.6 ± 1.4	5.9 ± 2.3	4.9 ± 1.8	2.68/0.07	

**Table 5 jcm-12-03707-t005:** Between group comparisons of the entire sample of the kyphotic group vs. normal group for sensorimotor control and CVA outcomes.

Variables	Kyphotic Group	Normal Group	Cohen’s d Effect Size	*p* Value [95% CI]
CVA (°)	41 ± 5	53 ± 4	2.65	<0.001 * [10.6, 13.4]
Smooth pursuit neck torsion test (% error)	0.41 ± 0.17	0.31 ± 0.14	0.6	<0.001 * [−0.15, −0.05]
** Overall stability index (refer to methods)	0.62 ± 0.2	0.42 ± 0.1	1.26	<0.001 * [−0.05, −0.14]
Head repositioning accuracy (°) Right	4.0 ± 1.5	3.0 ± 1.2	0.74	<0.001 * [−0.57, −1.42]
Head repositioning accuracy (°) Left	4.3 ± 1.8	3.3 ± 1.5	0.6	<0.001 * [−0.45, −1.58]
Sympathetic skin resistance Amplitude	2.9 ± 0.9	2.1 ± 0.7	0.87	<0.001 * [−0.54, −1.05]
Sympathetic skin resistance Latency	1.2 ± 0.4	1.3 ± 0.3	0.2	0.07 [−0.01, 0.21]

* Denotes statistically significant differences. ** These indices are calculated according to the degree of platform oscillation; smaller values indicate a good stability level of the participants. CVA = craniovertebral angle. All values are expressed as means ± standard deviation. CI [] = 95% confidence interval.

**Table 6 jcm-12-03707-t006:** Results of one-way-ANOVA and post hoc (Tukey) test. * = statistically significant.

Variables	Postural Kyphosis N = 35	Scheuermann’s Kyphosis N = 45	Normal Group N = 80	F-Value/ *p*-Value	Post Hoc
CVA (°)	44 ± 4	38.5 ± 4.5	53 ± 4	187.4/ <0.001 *	Group 1 vs. Group 2: Diff = −5.5, 95% CI = −8.58 to −2.4, *p* = 0.0002 * Group 1 vs. Group 3: Diff = 9, 95% CI = 5.7 to 12.27, *p* < 0.001 * Group 2 vs. Group 3: Diff = 14.5, 95% CI = 11.3 to 17.6, *p* < 0.001 *
Smooth pursuit neck torsion test (% error)	0.34 ± 0.13	0.48 ± 0.18	0.31 ± 0.14	19.1/<0.001 *	Group 1 vs. Group 2: Diff = 0.14, 95% CI = 0.059 to 0.22, *p* = 0.0002 * Group 1 vs. Group 3: Diff = −0.03, 95% CI = −0.10 to 0.04, *p* = 0.5 Group 2 vs. Group 3: Diff = −0.17, 95% CI = −0.24 to −0.10, *p* < 0.001 *
** Overall stability index (refer to methods)	0.56 ± 0.2	0.68 ± 0.3	0.42 ± 0.1	25.7/<0.001 *	Group 1 vs. Group 2: Diff = 0.12, 95% CI = 0.015 to 0.23, *p* = 0.02 * Group 1 vs. Group 3: Diff = −0.14, 95% CI = −0.23 to −0.045, *p* = 0.0017 * Group 2 vs. Group 3: Diff = −0.26, 95% CI = −0.35 to −0.17, *p* < 0.001 *
Head repositioning accuracy (°) Right	3 ± 0.7	4.8 ± 1.6	3.0 ± 1.2	33.84/ <0.001 *	Group 1 vs. Group 2: Diff = 1.8, 95% CI = 1.14 to 2.5, *p* < 0.001 * Group 1 vs. Group 3: Diff = 0.0, 95% CI = −0.59 to 0.59, *p* = 0.99 Group 2 vs. Group 3: Diff = −1.8, 95% CI = −2.34 to −1.25, *p* < 0.001 *
Head repositioning accuracy (°) Left	3.8 ± 2	4.7 ± 1.6	3.3 ± 1.5	10.39/0.04 *	Group 1 vs. Group 2: Diff = 0.9, 95% CI = 0.02 to 1.77, *p* = 0.04 * Group 1 vs. Group 3: Diff = −0.5, 95% CI = −1.29 to 0.29, *p* = 0.29 Group 2 vs. Group 3: Diff = −1.4, 95% CI = −2.12 to −0.67, *p* < 0.001 *
Sympathetic skin resistance Amplitude	2.4 ± 0.6	3.3 ± 1	2.1 ± 0.7	34.68/<0.001 *	Group 1 vs. Group 2: Diff = 0.9, 95% CI = 0.48 to 1.31, *p* < 0.001 * Group 1 vs. Group 3: Diff = −0.3, 95% CI = −0.67 to 0.07, *p* = 0.14 Group 2 vs. Group 3: Diff = −1.2, 95% CI = −1.54 to −0.85, *p* < 0.001 *
Sympathetic skin resistance Latency	1.3 ± 0.3	1.2 ± 0.5	1.3 ± 0.3	1.19/0.3	NA

* Denotes statistically significant differences. ** These indices are calculated according to the degree of platform oscillation; smaller values indicate a good stability level of the participants. CVA = craniovertebral angle. All values are expressed as means ± standard deviation.

**Table 7 jcm-12-03707-t007:** Correlations (Pearson’s *r*) between the postural kyphosis, the Scheuermann’s kyphosis, the normal group, and the entire sample for all measured outcomes.

Correlation between Variables	Postural Kyphosis r (*p* Value) N = 35	Scheuermann’s Kyphosis r (*p* Value) N = 45	Normal Group r (*p* Value) N = 80	Entire Sample r (*p* Value) N = 160
CVA	−0.7 (<0.001)	−0.6 (<0.001)	−0.51 (<0.001)	−0.61 (<0.001)
NDI	0.58 (<0.001)	0.50 (<0.001)	0.51 (<0.001)	0.67 (<0.001)
Pain intensity (NRS)	0.5 (<0.001)	0.35 (0.03)	0.34 (0.043)	0.53 (<0.001)
Smooth pursuit neck torsion test	0.54 (<0.001)	0.50 (<0.001)	0.50 (<0.001)	0.58 (<0.001)
Overall stability index	0.61 (<0.001)	0.49 (<0.001)	0.52 (<0.001)	0.59 (<0.001)
Head repositioning accuracy (Right)	0.7 (<0.001)	0.54 (<0.001)	0.61 (<0.001)	0.74 (<0.001)
Head repositioning accuracy (Left)	0.67 (<0.001)	0.52 (<0.001)	0.61 (<0.001)	0.71 (<0.001)
Sympathetic skin resistance amplitude	0.7 (<0.001)	0.56 (<0.001)	0.61 (<0.001)	0.69 (<0.001)
Sympathetic skin resistance latency	−0.2 (0.05)	−0.5 (<0.001)	−0.36 (<0.001)	−0.49 (<0.001)

CVA = Craniovertebral angle; NDI = neck disability index; NRS = numerical rating scale.

## Data Availability

Data supporting reported results can be ascertained by emailing the lead author of this study: Professor Ibrahim Moustafa at iabuamr@sharjah.ac.ae.

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
