# Peer review of "Is Thoracic Kyphosis Relevant to Pain, Autonomic Nervous System Function, Disability, and Cervical Sensorimotor Control in Patients with Chronic Nonspecific Neck Pain?"

_jcm, 2023, doi:10.3390/jcm12113707_

Round 1

Reviewer 1 Report (New Reviewer)

This paper described that the sensorimotor control, disability, and autonomic nervous system function of patients with chronic nonspecific neck pain and thoracic kyphosis is distinctly different compared to those patients with normal thoracic alignment. The authors collected the data from 80 participants with a defined hyper-kyphosis (> 55°) and 80 matched participants with normal thoracic kyphosis (< 55°). ALL of the participants neck pain.

The reviewer read this paper with great interest.

First of all, the reviewer is wondering how about the data of the normal control without neck pain. This sample should be the real normal control. These data should be corrected and shown.

This study does not include radiological analysis. The reviewer has a concern for the accuracy of the measurement. For example, the measurement of craniovertebral angle (CVA) is shown in Figure 3. However, this measurement should be altered the mass of the fat tissue in the neck. Did the authors compare the data according to the body mass index or body weight? This might not be accurate, the reviewer is afraid.

Although the authors stated that “We separated the hyper-kyphotic participants into two groups: 35 postural kyphosis and 45 Scheuermann’s kyphosis categories. ” How did the authors diagnose postural kyphosis and Scheuermann’s kyphosis. This diagnosis might be done by radiological examination.

What is the clinical relevance in this study? This point is very important. The authors should describe the clinical relevance in this study clearly.

The reference is not acceptable in the present form. There are numerous inappropriate capital letters in the title. These are very strange.

Almost OK except for the description of the reference

Author Response

Thank you for your detailed and thorough review of our manuscript. We believe it is much better now.

Reviewer #1 Comments:

  • General Overview: This paper described that the sensorimotor control, disability, and autonomic nervous system function of patients with chronic nonspecific neck pain and thoracic kyphosis is distinctly different compared to those patients with normal thoracic alignment. The authors collected the data from 80 participants with a defined hyper-kyphosis (> 55°) and 80 matched participants with normal thoracic kyphosis (< 55°). ALL of the participants neck pain.

  1. First of all, the reviewer is wondering how about the data of the normal control without neck pain. This sample should be the real normal control. These data should be corrected and shown.

Author Reply: We did not have a normal control group without neck pain. We included only participants with chronic non-specific neck pain who either had a normal thoracic kyphosis or who had hyper-thoracic kyphosis. Thus, our study design did not include the group that the reviewer is referring too. We have added this as a limitation in our discussion in lines 642-644 in the discussion. See highlighted section.

  1. This study does not include radiological analysis. The reviewer has a concern for the accuracy of the measurement. For example, the measurement of craniovertebral angle (CVA) is shown in Figure 3. However, this measurement should be altered the mass of the fat tissue in the neck. Did the authors compare the data according to the body mass index or body weight? This might not be accurate, the reviewer is afraid.

Author Reply: Our study did include a radiographic assessment of the group of 80 participants with thoracic hyper-kyphosis. This was necessary to rule out serious spine pathology and deformity and to identify the type of hyper-kyphosis that the hyper-kyphotic participant had. This has been highlighted for the reviewer on lines 120-122 of the methods section under 2.1 Participant inclusion.  Second, we previously had included a limitation to our study of lack of radiography. We have highlighted this for the reviewers awareness in the discussion -limitation section, 635-642.

Additionally, we have reported in the methods section previously and in the limitation sections that the measurements we have used for thoracic kyphosis and for the CVA are both valid and reliable. See reviewer references 14,36,37 for the CVA.

Lastly, our participants were matched for sex and size so this was a variable accounted for and did not alter our results.

  1. Although the authors stated that “We separated the hyper-kyphotic participants into two groups: 35 postural kyphosis and 45 Scheuermann’s kyphosis categories. ” How did the authors diagnose postural kyphosis and Scheuermann’s kyphosis. This diagnosis might be done by radiological examination.

Author Reply: This was previously presented in our original manuscript in section 2.1 Participant inclusion and exclusion criteria. We have added to this and clarified it on lines 120-124 in our methods section

  1. What is the clinical relevance in this study? This point is very important. The authors should describe the clinical relevance in this study clearly.

Author Reply: We appreciate this comment. We have added a clinical relevance paragraph near the end of our Discussion right before our limitations section. We have additionally added one more reference as a result. Lines 606-630 and reference 74.

  1. The reference is not acceptable in the present form. There are numerous inappropriate capital letters in the title. These are very strange.

Author Reply: We have corrected the references. Thank you.

Reviewer 2 Report (New Reviewer)

The overall writing of this manuscript is clear. The clinical implication of this study is well justified and important. However, I have some concerns about the study method, which may influence the interpretation of the results.

Line 57-58: As stated here, neck disorder and thoracic hyper-kyphosis are more prevalent in the elderly. It is surprising to see that the mean age of the participants in this study is around 25.

Line 122: How was Scheuermann’s kyphosis diagnosed? What criteria were used to classify the participants as mild or moderate Scheuermann’s kyphosis?

Line 168: It is a bit confused here that a 4D system (Line 126) was used to analyze 3D body posture. The authors should justify why they chose formetric measurement over radiographic measurement instead of repeatedly discussed the over-estimation issue when comparing formetric and radiographic measurement and the cutoff value for thoracic hyperkyphosis.

Line 182: Was the image taken with the participants wearing shoes? Footwear and ankle posture may also influence posture.

Line 222: Please add the product name and other relevant information.

Line 223: As stated in the Introduction, thoracic posture may influence cervical movement. Please explain how upright sitting was defined during testing.

Line 234:  Please add the product name and other relevant information.

Line 255: Please justify for the choice of sympathetic skin response for autonomic nervous system function, considering that the authors emphasized the importance of the 10th cranial nerve. Why did not you measure heart rate variability?

Line 280: Which outcome measure(s) was/were used for sample size calculation?

Line 281: This is a case control study not an interventional study. I don’t understand why the authors needed to consider possible drop-out. Were outcome measures taken on separate days?

Line 283: I have great concern about multiple comparisons using the Student’s t-test. The readers may like to know where there is a statistically significant difference between the postural kyphosis group and the normal kyphosis group and between the Scheuermann’s kyphosis group and the normal kyphosis group. One-way ANOVA may be more appropriate.

Line 315: The postural kyphotic group in Tables 1 and 2 are different. Please use different names to avoid confusion. Table 1 has an extra blank row and the row with “smoking” should be placed in the middle.

Line 407, 412: Data in the Scheuermann’s kyphosis group did not include those with more severe Scheuermann’s kyphosis. A weaker correlation in the SSR amplitude and other outcome measures may be simply the result of more cluster data. On the other hand, the participants in the postural kyphosis group are likely more heterogeneous. The stronger correlation may be simply the result of more scattered data. The current interpretation may be misleading.

Line 424-426: Is this sentence “The normal kyphosis group had the greatest CVA….” based on two t-tests comparing the difference between the normal kyphosis group and the entire kyphotic group and between the postural kyphosis group? You can’t draw such an inference from two unpaired t-tests.

Line 453: The mean age of participants in two cited references is about 50 years, much older than the participants in this study. As I commented above, it is surprising to see this age group having both chronic nonspecific neck pain and thoracic kyphosis and seeking medical treatment in clinical setting.

Line 514-515: The conclusion of this SR is based on three highly heterogenous studies. The certainty of evidence is not high.

Line 517-518: Please explain in more details.

Line 556-558: Missing citation.

Author Response

Thank you for your detailed review of our original manuscript. We believe we have revised according to all your critical assessments and believe our manuscript is much better.

Reviewer #2 Comments:

  • General Comments: The overall writing of this manuscript is clear. The clinical implication of this study is well justified and important. However, I have some concerns about the study method, which may influence the interpretation of the results.

  1. Line 57-58: As stated here, neck disorder and thoracic hyper-kyphosis are more prevalent in the elderly. It is surprising to see that the mean age of the participants in this study is around 25.

Author Reply: We agree with this statement and we did reference that. However, we also (on lines 58-63) brought up the fact that while young adults do not have the highest prevalence of neck disorders, the data suggests that it is still quite high for young adults. For example, according to several studies, neck pain is one of the most prevalent musculoskeletal conditions worldwide, with a reported 12-month prevalence in young adults ranging from 42 to 67%. We have already presented this information to the Introduction and to our references. See highlights in Introduction, 1st paragraph and references 9-11.

  • Garni AD, Al-Saran Y, Al-Moawi A, Bin Dous A, Al-Ahaideb A, Kachanathu SJ. The Prevalence of and Factors Associated with Neck, Shoulder, and Low-Back Pains among Medical Students at University Hospitals in Central Saudi Arabia. Pain Res Treat. 2017;2017:1235706. [PMC free article] [PubMed] [Google Scholar]
  • Alshagga MA, Nimer AR, Yan LP, Ibrahim IA, Al-Ghamdi SS, Radman Al-Dubai SA. Prevalence and factors associated with neck, shoulder and low back pains among medical students in a Malaysian medical college. BMC Res Notes. 2013;6:244. [PMC free article] [PubMed] [Google Scholar]
  • Almhdawi KA, Mathiowetz V, Al-Hourani Z, Khader Y, Kanaan SF, Alhasan M. Musculoskeletal pain symptoms among allied health professions' students: prevalence rates and associated factors. J Back Musculoskelet Rehabil. 2017;30(6):1291–1301. [PubMed] [Google Scholar]

  1. Line 122: How was Scheuermann’s kyphosis diagnosed? What criteria were used to classify the participants as mild or moderate Scheuermann’s kyphosis?

Author Reply: All 80 participants in the hyper-kyphotic group underwent radiological examination to identify the possible cause of their hyper-kyphosis. This is standard in a younger population to determine the mechanisms for this. We have clarified this on lines 108-109 in the materials and methods 1st paragraph and in section 2.1. Inclusion criteria lines 120-124. Mild to moderate was the classification given by the Orthopedic Surgeon based on wedging deformity and kyphosis magnitude.

  1. Line 168: It is a bit confused here that a 4D system (Line 126) was used to analyze 3D body posture. The authors should justify why they chose formetric measurement over radiographic measurement instead of repeatedly discussed the over-estimation issue when comparing formetric and radiographic measurement and the cutoff value for thoracic hyperkyphosis.

Author Reply: We have clarified the use of the 4D system on line 131-132. The 4D system allows for a time of capture variable of 60 seconds to account for any sagittal sway or shifting.

We have clarified that Participants with normal kyphosis did not receive thoracic spine x-ray imaging as there was no clinical rationale for imaging in these participants, thus, an external measurement of thoracic kyphosis was chosen to make comparisons in all participants. See lines 126-129. We argue that it is very relevant and important to present the reliability and validity of the Diers formetric system versus conventional radiography so that the application and limitations of our results can be properly understood. Our presentation of this information in the methhods and discussion is warranted.

  1. Line 182: Was the image taken with the participants wearing shoes? Footwear and ankle posture may also influence posture.

Author Reply: Thank you for this clarification. We agree that footwear would alter the results which is why all participants were barefoot during examination. We have added this on line 183.

  1. Line 222: Please add the product name and other relevant information.

Author Reply: On lines 229-230 we have added the CROM product information and manufacturer and purchase location.

  1. Line 223: As stated in the Introduction, thoracic posture may influence cervical movement. Please explain how upright sitting was defined during testing.

Author Reply: We have highlighted the protocol we used and clarified the thoracic posture and instructions to the participants. We followed the published protocol of Loudon et al. Our reference 40. See lines 231-235.

  1. Line 234: Please add the product name and other relevant information for the SPNT Equipment.

Author Reply:  This information has been added to the methods. Lines 254-256.   The videonystagmography system VisualEyes™ 525 by Interacoustics A/S in Denmark was utilized to conduct the SPNT test.

  1. Line 255: Please justify for the choice of sympathetic skin response for autonomic nervous system function, considering that the authors emphasized the importance of the 10th cranial nerve. Why did not you measure heart rate variability?

Author Reply: We have added some of the information below to our discussion section under the SSR topic. See highlights lines 578-593 and new references 71-72.

We appreciate the reviewer's comment on the choice of sympathetic skin response (SSR) as an indicator for autonomic nervous system (ANS) function and the absence of heart rate variability (HRV) measurement in our study. We agree that HRV is a commonly used and standardized method for assessing ANS function, and that it provides separate metrics for sympathetic and parasympathetic functions through the low-frequency (LF) and high-frequency (HF) spectral components of HRV. However, recent studies have shown that the traditional HRV framework established in the 1980s has limitations in dealing with the vast evidence accumulated over the past half-century. As pointed out by Hayano and Yuda (2019), using HRV without criticism may lead to incorrect conclusions or judgments. Moreover, a study by Ke et al. (2017) has shown that both SSR and HRV parameters are sensitive in determining ANS dysfunction. Therefore, we chose SSR as an alternative measure for ANS function in our study.

We acknowledge that HRV may provide additional information about ANS function, and we will consider including HRV measurement in our future studies. However, our current findings using SSR highlight the potential clinical value of this measure in assessing ANS dysfunction.

  • Hayano, Junichiro, and Emi Yuda. 2019. “Pitfalls of Assessment of Autonomic Function by Heart Rate Variability.” Journal of physiological anthropology 38(1). https://pubmed.ncbi.nlm.nih.gov/30867063/ .
  • Ke, Jiang Qiong et al. 2017. “Sympathetic Skin Response and Heart Rate Variability in Predicting Autonomic Disorders in Patients with Parkinson Disease.” Medicine 96(18). https://pubmed.ncbi.nlm.nih.gov/28471954/.

  1. Line 280: Which outcome measure(s) was/were used for sample size calculation?

Author Reply:   Pain was used as the outcome measure for this calculation. We have indicated this on line number 295

  1. Line 281: This is a case control study not an interventional study. I don’t understand why the authors needed to consider possible drop-out. Were outcome measures taken on separate days?

Author Reply: We unintentionally made this statement as we have done several RCT’s too and this was an oversight. We have removed this statement and all outcome measures were taken on the same day for each given participant. See corrected lines 295-297.

  1. Line 283: I have great concern about multiple comparisons using the Student’s t-test. The readers may like to know where there is a statistically significant difference between the postural kyphosis group and the normal kyphosis group and between the Scheuermann’s kyphosis group and the normal kyphosis group. One-way ANOVA may be more appropriate.

Author Reply: We have redone our statistical analysis in the methods section (lines 304-308) using one-way ANOVA and a Post hoc Tukey’s analysis was performed to determine differences when the ANOVA identified a significant difference.

We further revised our Results section to report this data. See new information in Table 2 and Table 4 and Table 6.

  1. Line 315: The postural kyphotic group in Tables 1 and 2 are different. Please use different names to avoid confusion. Table 1 has an extra blank row and the row with “smoking” should be placed in the middle.

Author Reply: We have revised these groups in the Tables as suggested. And we have removed the blank row. See revised Table 1-2.

  1. Line 407, 412: Data in the Scheuermann’s kyphosis group did not include those with more severe Scheuermann’s kyphosis. A weaker correlation in the SSR amplitude and other outcome measures may be simply the result of more cluster data. On the other hand, the participants in the postural kyphosis group are likely more heterogeneous. The stronger correlation may be simply the result of more scattered data. The current interpretation may be misleading.

Author Reply  

We appreciate your concern about the homogeneity of variance assumption in our study. We used the Levene statistic to assess this assumption in our analysis, and based on the results, we proceeded with our analysis under the assumption of homogeneity of variance. See our methods section lines 300-301 under Statistical analysis.

  1. Line 424-426: Is this sentence “The normal kyphosis group had the greatest CVA….” based on two t-tests comparing the difference between the normal kyphosis group and the entire kyphotic group and between the postural kyphosis group? You can’t draw such an inference from two unpaired t-tests.

Author Reply   Thank you for pointing out our error, we have implemented the one way analysis of variance as suggested above in our methods and our results.

  1. Line 453: The mean age of participants in two cited references is about 50 years, much older than the participants in this study. As I commented above, it is surprising to see this age group having both chronic nonspecific neck pain and thoracic kyphosis and seeking medical treatment in clinical setting.

Author Reply: We disagree with this statement. While it is true that we cited articles on older populations having a greater frequency of chronic nonspecific neck pain, we also cited relevant data showing that younger populations have a relatively high prevalence of chronic nonspecific neck pain too. We repeat our reply to comment #1 above:

“For example, according to several studies, neck pain is one of the most prevalent musculoskeletal conditions worldwide, with a reported 12-month prevalence in young adults ranging from 42 to 67%. We have already presented this information to the Introduction and to our references. See highlights in Introduction, 1st paragraph and references 9-11.

  • Garni AD, Al-Saran Y, Al-Moawi A, Bin Dous A, Al-Ahaideb A, Kachanathu SJ. The Prevalence of and Factors Associated with Neck, Shoulder, and Low-Back Pains among Medical Students at University Hospitals in Central Saudi Arabia. Pain Res Treat. 2017;2017:1235706. [PMC free article] [PubMed] [Google Scholar]
  • Alshagga MA, Nimer AR, Yan LP, Ibrahim IA, Al-Ghamdi SS, Radman Al-Dubai SA. Prevalence and factors associated with neck, shoulder and low back pains among medical students in a Malaysian medical college. BMC Res Notes. 2013;6:244. [PMC free article] [PubMed] [Google Scholar]
  • Almhdawi KA, Mathiowetz V, Al-Hourani Z, Khader Y, Kanaan SF, Alhasan M. Musculoskeletal pain symptoms among allied health professions' students: prevalence rates and associated factors. J Back Musculoskelet Rehabil. 2017;30(6):1291–1301. [PubMed] [Google Scholar]”

  1. Line 514-515: The conclusion of this SR is based on three highly heterogenous studies. The certainty of evidence is not high.

Author Reply: We have added another recent systematic review with meta analysis to strengthen this statement: See new reference 68:

Rani B, Paul A, Chauhan A, Pradhan P, Dhillon MS. Is Neck Pain Related to Sagittal Head and Neck Posture?: A Systematic Review and Meta-analysis. Indian J Orthop. 2023 Jan 18;57(3):371-403. doi: 10.1007/s43465-023-00820-x. PMID: 36825268; PMCID: PMC9941407.

  1. Line 517-518: Please explain in more details.

Author Reply: We have clarified this section. See lines 553-557.

  1. Line 556-558: Missing citation.

Author Reply: We have added citations 73-74 herein. Line 613.

Round 2

Reviewer 1 Report (New Reviewer)

The paper has been improved.

This manuscript is a resubmission of an earlier submission. The following is a list of the peer review reports and author responses from that submission.

Round 1

Reviewer 1 Report

I appreciate the opportunity to review this study. The authors have conducted a study aiming to s to investigate the changes in sensorimotor control, disability, and autonomic dysfunction in chronic nonspecific neck patients with thoracic hyperkyphosis compared to strictly matched control participants with a normal thoracic sagittal alignment.

The manuscript is well-written and detailed; it is apparent great time and effort went into the prepared manuscript. I have some comments about the manuscript which I will list below.

- Page 2, lines 71-74: Authors described “the vast amount of knowledge” but cited only one study (19);

- I would suggest to the authors include some references about the relationship between pain and posture as we have some studies in this area. In the same way, between disability and thoracic posture as well.

- As a cross-sectional study, I would recommend using the STROBE guidelines for reporting observational studies.

- What was the instruction for the participants before the measurements of thoracic kyphosis and CVA? In addition, the authors presented the reliability of thoracic kyphosis measurement, but what is the reliability of the CVA?

- What is the clinical relevance of the sensorimotor control variables? A change in posture might be responsible for these differences between groups.

- Page 9, lines 253-255: “We found moderate positive correlation between the kyphotic angle and SSR amplitude (r = 0.61), which means if the kyphotic angles increase, the SSR amplitudes will increase”. No, you cannot say that. As a cross-sectional study, you cannot state a cause-effect relationship. Please, review these statements throughout the results and discussion sections.  

Reviewer 2 Report

Dear authors,

Thank you for the opportunity to review the manuscript titled: “Is thoracic posture relevant to pain, autonomic nervous system function, disability, and cervical sensorimotor control in patients with Chronic nonspecific neck pain? A Cross Sectional Study”. I think the study has good findings related to kyphotic angles and have relatively big sample size. However, I found some needs for clarifying wording, revising analysis, and providing more rationale for the measures that authors used. Particularly, too many acronyms and current tables/figures were difficult to follow.

Title

The capitalization of title is inconsistent. I will recommend changing the title to: Is thoracic posture relevant to pain, autonomic nervous system function, disability, and cervical sensorimotor control in patients with chronic nonspecific neck pain?

Overall comment

References in the order of three consecutive numbers are inconsistently written. For example, line 56 and 68 should be [5-7] and [13-15], respectively, like you did for line 76: “[5-8]”.

Abstract

Please check JCM guidelines. I don’t think headings are needed.

Too many acronyms are used in this short abstract. I would avoid using too many abbreviation. I also think it is not clear what your purpose is, which makes the reader confused when reading the results.

Line 16: The sentence sounds a bit awkward. Is there a growing interest or growing concern?

Line 17: outcome of healthcare is such a broad term. Please provide specific outcome

Line 18: defined HTK? Is it influence of HTK? I don’t think defined is needed here.

Line 20: definite HTK? I don’t understand what definite implies.

Line 22: Acronym NRS is not used after this in the abstract. So I will remove NRS

Line 24: variables? Or “Sensorimotor control were measured by..”?

Line 26: what tests did you use to compare the differences?

Line 27: the relationship “among”?

Line 27-28: you say some variables in a broader term (like sensorimotor control), but you say specific measures for CVA, which is not consistent. Please choose general or specific measures that you’ve sough the correlation and provide it with consistency.

Line 27: From this correlation analysis, Did you combine the two groups of HTK and NTK for correlation? I am not quite understanding your purpose of correlation here.

Line33: “(p<0.001), but”

1.Introduction

I didn’t find clear rationale for you doing all measures and seeing independent correlation with kyphotic angle.

First paragraph: I think you have needed information there, but the information is bit scattered. You mentioned Thoracic spine way to early in the context and go back and forth with the biomechanical factors contributing to neck pain. I will convey more bigger picture first and then end up with thoracic spine dysfunction (which is supported by your next paragraph)

Fourth paragraph: I also think you have many information here, but seems not aligned in a logical order. Can you list things in order of known à unknown à and what needs to be addressed?

Line 79: I don’t think you are looking at the “changes” since this is cross-sectional study? “differences” might be more accurate?

2. Materials and Methods

For participants, did you not have age criteria for the hyper-kyphosis? That means children can participate, too? Please include justification of why you didn’t set age criteria as children, young adults and older adult’s kyphosis is different.

Line 82-84: I see your hypothesis only is implying comparisons between groups. You stated in your abstract that you are looking at some correlation analyses. I think you need to add those

Line 86-88: can you add the definition of thoracic hyper-kyphosis and normal thoracic kyphosis here? Your next paragraph seems more reasonable to come before line 86.

Line 97: what is ICT-ITL? No explanation here.

Line 106: I suggest correcting Figure 2 resolution. Please add all abbreviation of ICT, ITL, ILS, KA, LA, VP, and DM in the figure caption.

Line 125: Figure 3 seems unnecessarily big.

Line 206: When you mention correlation, are you putting all group data (N=160) together? This is not clear and confusing throughout. I don’t see the point of dividing the two groups for correlation since both groups have same direction in the correlation. Kyphotic angle is continuous variable (which divides the group), so looking at the correlation as a whole group (N=160) can strengthen your correlation coefficient. Have you checked this?

3. Results

Line 210: Results header is in weird spot.

Line 215: Please provide test results (p-value) for the table 1. You mentioned Age and sex were matched between kyphotic and normal group, but I don’t understand how sex and age could be different between the kyphotic and control group. And I don’t think the group should be called control as demographic is not matched.

Table 1. You included Kyphotic angle and CVA in demographics table. You mentioned there is no group differences in line 214. You also included CVA in Table 3, which is overlapping.

Line 225: Figure 4. I actually don’t see the point of only adding this figure. I will include all your variables as figure or just have table as you are providing redundant information. If you decide to put figure 4, please include individual dots for the figure. And add horizontal line in 55 deg, which is the criteria for dividing the group in your study.

Line 243: Table 3. I appreciate you providing the acronym in the caption, but this table is very difficult to read. Can you just write the full name in the variables box? Why is CVA balded when other results are significant too?

Line 261: Table 4. CVA should be -0.51? typo found. I also think p-values for only <0.001 should be reported with “<”, not when it is 0.043 and 0.040. you should remove the symbol. Since your kyphotic angle is correlated with multiple variables, I would recommend using multiple regression than reporting all independent correlation between two variables.

Line 243: Please provide units for all variables. As you indicated in line 284, I recommend including minimal detectable change or minimal clinically important difference as 1.0 seems not like a big deal (although you see group difference).

Line 271: Figure 5. I don’t see your point of adding this figure as it is redundant. And your unit seems like a letter ‘o’. you have inconsistent x-axis that includes “G” as group.

4. Discussion

Line 288-305: I wonder why you chose 55 deg when there are so many cut-off for hyperkyphosis. You mention the group average as a whole group, which is confusing why you would mention this as a whole group when you provided the separate mean for hyper- and normal kyphosis groups?

Line 408: References format is not following JCM guidelines.

Round 2

Reviewer 1 Report

I appreciate the opportunity to review the revised version of this study. I commend the authors on their work; however, I still have some concerns about the methods and results sections that I would to comment on below.

1)      I am still concerned about the method used to measure thoracic kyphosis, for instance, when the authors report that “the participant was correctly positioned in the neutral posture position”, how was the participant instructed to stay in a neutral position? What would be a neutral position? And finally, why would you measure the thoracic kyphosis in a neutral position?

2)      I understand how the sensorimotor control variables were measured; however, I still do not understand their clinical relevance. Clinical relevance is related to the relationship between measurements and clinical aspects important for patients. For instance, what is the relationship of sensorimotor control with pain and disability?

3)      Finally, we have several studies showing no relationship between posture and pain in subjects with spinal pain. In this way, I would recommend to the authors review the introduction section adding some evidence on the importance of this investigation.

4)      Please, consider reading some of this literature that may help to discuss your results (DOI: 10.1007/s12178-019-09594-y, DOI: 10.1080/08990220.2018.1475352, DOI: 10.2522/ptj.20150241).

Reviewer 2 Report

Abstract - doesn't read well.

Line 19: driving factors

- from the first three sentence you seem to investigate the contributing factors to neck pain, disability, and function. Because you are suggesting those are unknown, but your study analysis and design does not address the gap you mentioned. In your purpose, you say you want to compare differences in sensorimotor etc. I would reflect that purpose in the beginning of your abstract. 

Line 32: disability - what do you mean by disability?

Introduction

First paragraph - improved

4th paragraph - ASD doesn't show up often. please avoid using unnecessary acronym. I am not sure line 80-83 flows well within the paragraph.

I still think the flow of the introduction does not help reader understand why you are conducting comparisons when it's actually correlation that you are trying to answer.

Methods

Line 116: What is the error of measurement for 4D formetric system? If you had 53 or 54 deg individual that were included in the control group, does 4D formetric system have less than 1 deg of measurement error?

Figure 4 - still says control

Table 4 - NDI Entire sample - I think you should be 0.67 not -.67

Table 4 - the correlation of entire sample shows better overall correlation than separating the group because your group separation is linear. I don't think you added discussion about this, which should be addressed.

Figure 5 - still written as "Control"

Discussion

I think you should be reading your discussion and check to see if your abstract is conveying the same information.

If you separate the group of the two that are within a linear relationship, you will most obviously find a difference. This is not surprising .

Line 393-396: does not read well. Need English editing.

Line 422: you talk about CVA and neck pain - but you did not find pain difference between the groups. Can you give more explanation why your finding was different than Lau's?
